# The Growth Factor Release from a Platelet-Rich Plasma Preparation Is Influenced by the Onset of Guttate Psoriasis: A Case Report

Elisa Borsani [1,2,*] , Barbara Buffoli [1,2] , Francesca Bonomini [1,2] and Rita Rezzani [1,2]

1 Department of Clinical and Experimental Sciences, Division of Anatomy and Physiopathology, University of Brescia, 25123 Brescia, Italy; barbara.buffoli@unibs.it (B.B.); francesca.bonomini@unibs.it (F.B.); rita.rezzani@unibs.it (R.R.)
2 Interdepartmental University Center of Research "Adaption and Regeneration of Tissues and Organs-(ARTO)", University of Brescia, 25123 Brescia, Italy
* Correspondence: elisa.borsani@unibs.it

**Featured Application: This study evaluates a case of psoriasis presenting an imbalance of growth factor production from blood cells before the onset of the pathology, detected by ELISA. Our data could open a new hypothesis on the role of platelets in this pathology.**

**Abstract:** The involvement of platelets in immune and inflammatory processes is generally recognized; nevertheless, in psoriasis, their role is not clearly understood. We studied the in vitro growth factor release from a platelet-rich plasma preparation, the concentrated growth factors (CGF), in a case of a psoriasis subject three days before the onset of the papule. The CGF clots were incubated in a cell culture medium without growth supplements for 5 h and 1, 3, 6, 7, and 8 days, and the release kinetics of PDGF-AB, VEGF, TNF-α, and TGF-β1 were evaluated. The data, based on the results obtained during the case study, report a general increase in growth factor release in the psoriasis subject with respect to the healthy control, indicating an imbalance of growth factor production from blood cells. Although the results should be validated in the future, they show new aspects of this dermatological pathology, opening new possibilities both as the method of study, using CGF, and the involvement of platelets and growth factors in its development and maintenance.

**Keywords:** concentrated growth factors (CGF); platelet-rich plasma preparation; growth factors; psoriasis

## 1. Introduction

Psoriasis is a common autoimmune inflammatory skin disease with an incomplete understood etiology [1,2] in which genetic predisposition and environmental factors contribute to its manifestation in susceptible individuals [3]. The first most common type is named chronic plaque psoriasis (psoriasis vulgaris), characterized by red scaly plaques on the skin; the second most common type is named guttate psoriasis, characterized by small, separate, and red spots on the skin that number in the hundreds. In both cases, the deep red coloration is due to dermal angiogenesis, which allows for a greater influx of inflammatory cells into the skin. Currently, the involvement of platelets in inflammatory and immune processes is identified more than ever before [4–6]. Regarding platelet activation, numerous secretory mediators are released, and different kinds of adhesive and immune receptors are expressed on their membranes, leading to the initiation and modulation of inflammatory and immune responses [1]; nevertheless, they are able to synthesize the growth factors up to over 7 days after activation from accumulated mRNA [7]. The mean platelet volume (MPV), which is used as a marker of platelet activation [8,9], is increased during psoriasis, confirming the presence of larger, more reactive platelets resulting from heightened platelet

turnover [10]. However, the exact mechanisms of these associations, especially before the onset of the pathology, remain unclear.

Today, numerous systemic treatments specifically for severe types of these diseases are proposed to patients to improve their quality of life. In this context, the platelet-rich preparations represent a novel approach to tissue regeneration. They are characterized by a concentration of platelets above baseline normal count in a small volume of plasma [11] with different ratios of platelets, leukocytes, growth factors, and fibrin matrix [12,13]. Among them, in this study, we used the concentrated growth factors (CGF) [14,15] as a tool to study systemic pathologies. Our previous data [14] showed a specific kinetic of accumulation for each growth factor analyzed in vitro over 8 days, sustaining the ability of platelets to guarantee the release of the growth factors for several days after activation [7]. The leukocytes contribute to growth factor accumulation as reported for other platelet-rich preparations, e.g., [16].

This study aimed to describe a case of guttate psoriasis presenting an imbalance of growth factor production from blood cells before the onset of the pathology, detected by ELISA. Our data could open a new hypothesis on the role of platelets in psoriasis.

## 2. Materials and Methods

### 2.1. Participants

Two adult volunteers (V1 and V2), both Caucasian females, initially recruited to characterize the growth factor release of CGF, were considered in this study. One of them (V2) presented altered growth factor release in respect to the normal range [14], and thus, her medical recent history was investigated. The hematologic blood test (leukocytes, platelets, and erythrocytes count) was performed by S.T.E.M. laboratory (Brescia, Italy) before recruitment. The study was carried out according to the Helsinki declaration and written informed consent for the re-use of human bio-specimens in scientific research was obtained from both subjects enrolled in the study.

Exclusion criteria were the presence of any systemic disorder, smoking, active or chronic infection, non-steroidal anti-inflammatory drug use, and altered values of routine blood panel.

### 2.2. Case Reports

Healthy volunteer (V1): a 38-year-old healthy woman with no established dermatological conditions at the time of the blood collection.

Case report-guttate psoriasis (V2): a 37-year-old woman with no established dermatological conditions at the time of the blood collection. Nevertheless, after 3 days of blood collection, she noticed the generalized onset of small, drop-like, salmon-pink papules, sub-sequentially clinically diagnosed as guttate psoriasis.

### 2.3. Blood Collection

The whole blood was collected by arm venipuncture with a 21-gauge and always immediately processed. The experiments were performed using one CGF for each time point; thus, each volunteer had a blood collection for a total of 6 tubes for the evaluation of cumulative growth factor release.

### 2.4. CGF Preparation

The CGF was obtained following this protocol: 9 mL of blood was drawn in each sterile Vacuette tube (Greiner Bio-One, GmbH, Kremsmunster, Austria) silicon-coated as serum clot activator and then immediately centrifuged (30′ acceleration, 2′ 2700 rpm, 4′ 2400 rpm, 4′ 2700 rpm, 3′ 3000 rpm, and 33′ deceleration and stop) into a special centrifuge machine (Medifuge MF200, Silfradent Srl, Forlì, Italy). At the end of the process, the obtained solid CGF was isolated as previously described [14,17].

### 2.5. Cumulative Growth Factor Release

The CGFs were placed in 12-well plates (one in each well) with the addition of 1.6 mL of cell culture medium RPMI 1640 (Lonza, Verviers, Belgium) without growth supplements and then incubated at 37 °C for 5 h and 1, 3, 6, 7 and 8 days [18]. After each incubation period, the medium was collected and centrifuged at $400\times g$ for 10 min at room temperature. The supernatant was stored at −80 °C until analysis [18]. The quantification of growth factors (PDGF-AB, VEGF, TNF-$\alpha$, and TGF-$\beta$1) was performed using ELISA kits according to the manufacturer's protocol (R&D Systems Inc., Minneapolis, MN, USA). The total amount of growth factors in the medium from all time points was checked obtaining the kinetics of each growth factor analyzed.

### 3. Results

The data showed an anomalous release of growth factors in the V2 volunteer compared with healthy V1 (Figure 1).

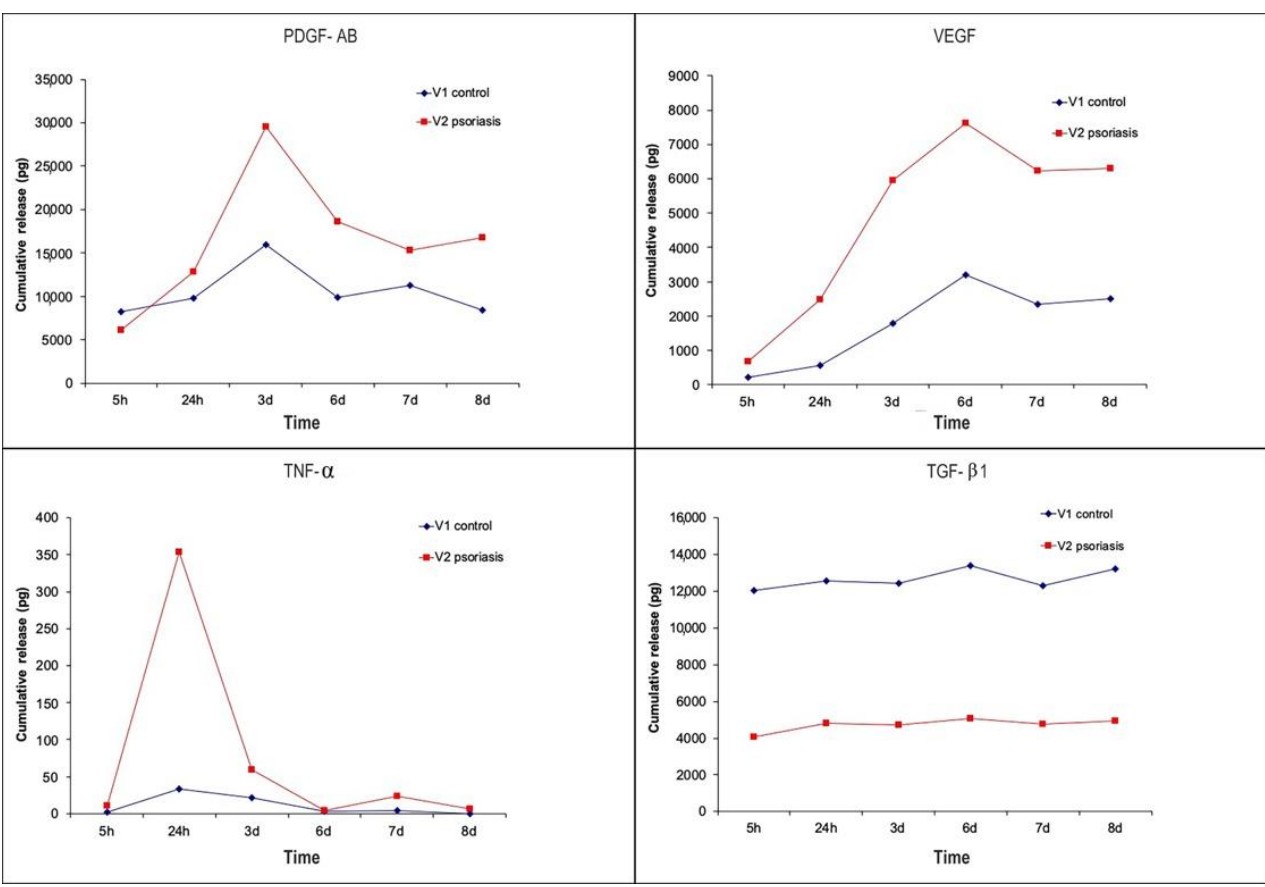

**Figure 1.** Cumulative growth factor released by CGF at 5 h (5 h), 1 day (24 h), 3 days (3 d), 6 days (6 d), 7 days (7 d), and 8 days (8 d) of culture. Each volunteer's values are plotted: healthy control (V1), volunteer 3 days before the onset of guttate psoriasis (V2). The values are reported as the total amount of growth factor released: PDGF-AB, VEGF, TNF-$\alpha$, TGF-$\beta$1.

Our results show that the growth factors released from CGF were altered before the onset of guttate psoriasis. Generally, it greatly increased with psoriasis compared with the control (Figure 2).

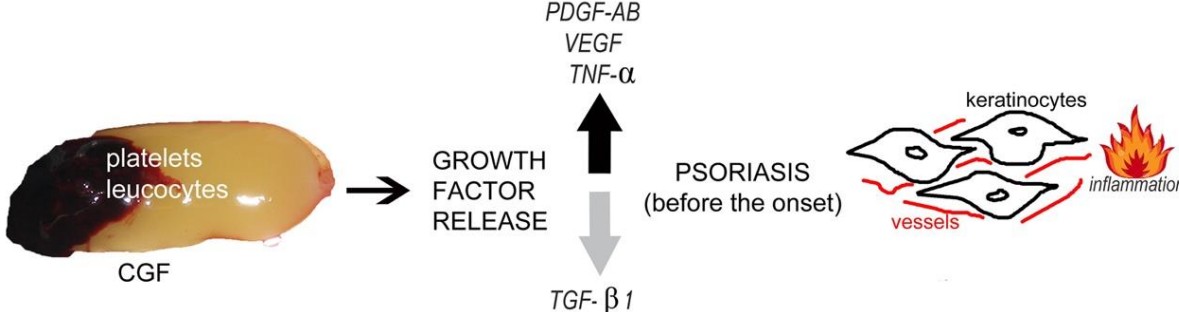

**Figure 2.** Schematic diagram of the potential influence of growth factors from blood cells on keratinocytes in psoriasis. Growth factors released from CGF are altered in the presence of skin pathology. Generally, it greatly increased with psoriasis and decreased with vitiligo compared with the control. We could suppose that these cells may have inherent aberrations that make them particularly vulnerable to extracellular insult; thus, an alteration in the supply of trophic factors concurs with the onset of the pathology.

*3.1. Healthy Case*

Healthy V1 presented a normal range release, as previously reported [14]. PDGF-AB had a constant accumulation reaching the top on the 3rd day; VEGF increased during the experiment period, reaching the maximum accumulation on the 6–8th day; TNF-α reached the maximum accumulation on the 1st day and then progressively decreased until the 8th day; TGF-β1 had a constant accumulation during all the experimental periods.

*3.2. Case Report–Guttate Psoriasis before the Onset*

V2 (psoriasis) presented a general increase in growth factors release, except for TGF-β1, which decreased, maintaining the same kinetic release of V1. In particular, the PDGF-AB began to increase at 24 h and remained high for the period checked; on the 3rd day, a peak was reported with a value about double the control. VEGF showed a doubled value concerning V1 at each time point. TNF-α was maximally released at 24 h, where the value was about 7-fold higher than the control. Regarding TGF-β1, it had a constant accumulation during the experimental period, but the trend was inverted compared with the other growth factor examined, and the release was reduced by approximately 1/3 compared with V1.

**4. Discussion**

New systematic treatment approaches for psoriasis are numerous with the aim to improve patients' quality of life; however, there is still a need for new therapeutic alternatives where these treatment modalities are not suitable. If the finding of this study could be confirmed by a wide clinical study, a new hypothesis could be made about the involvement of the platelets in the onset of this pathology, and the CGF could be a useful in vitro tool to evaluate the kinetic release of growth factors over time; moreover, it could simulate the psoriasis microenvironment in 2D and 3D human skin models.

Nevertheless, this study presents some limitations that must be considered in the comprehensive evaluation of the obtained results during the case study. The first limitation is the evaluation of only one psoriasis subject, which is a case report. Thus, the researchers want to address a new hypothesis on the onset of this pathology considering the data of this study, which should be validated in the future. The second limitation is the presence of leucocytes in CGF, which can secrete TNF-α, VEGF, TGF-β1, and PDGF-AB. Thus, the data obtained cannot be associated with only one cell type even if, in this study, based on our previous morphological research, the contribution of platelets in the growth factor production was expected to be important [14,19]. The quantitative count of the different cell types was hindered by the solid nature of CGF; the cells are entrapped in the fibrin network formed during the centrifugation without the aid of additives. Nevertheless, a study by Anitua et al. [16] evidenced a specific trend of VEGF release and an alteration of the fibrin

network in the presence of leukocytes. Two different platelet-rich plasma preparations have been considered: leucocyte free and leucocyte rich. The release of different growth factors was evaluated in vitro over 8 days, and the availability of VEGF greatly decreased after 3 days and was almost absent on the 7th day using the leucocyte-rich preparation. On the contrary, the VEGF kinetic release observed in our experiments on CGF was similar to the leucocyte-free preparation. Moreover, the fibrin network in the leukocyte-rich preparation resulted in a more heterogeneous and loose mesh, which was less evident in CGF during our previous studies [14].

The results obtained from V2 are valuable and important because they are usually difficult to obtain; they permit the description of the growth factor status in patients some days before symptoms onset. Our data showed a general increase in growth factors released that could sustain some hypotheses described in the scientific literature. Evidence for an in vivo platelet activation, which could contribute to the development of thrombotic events, has been established in psoriasis patients [4]. Recently, Takeshita et al. [20] demonstrated a higher presence in the blood of psoriasis patients of microparticles derived from platelets, endothelial cells, and monocytes/macrophages compared with controls.

Some markers of platelet activation, such as mean platelet volume (MPV), spontaneous platelet hyperaggregability, platelet factor 4, and plasma levels of β-thromboglobulin, were significantly higher in psoriasis patients. In contrast, platelet count and circulating platelet aggregates were normal. In addition, about a decade ago, some studies [21,22] suggested the central role of cytokines expressed in psoriasis skin during the disease process; the interplay among them could explain most of the clinical features of psoriasis, such as the hyperproliferation of keratinocytes, increased neovascularization, and inflammation. Moreover, in 2009, Gisondi et al. [23] hypothesized that the deranged hemostatic balance might be mainly sustained by platelet hyperactivity. In this context, a clinical relapse of this preliminary report could introduce the idea of a potential preventive intervention that could be provided to patients before psoriasis onset with the aim to modulate growth factor secretion. Some target pharmacological therapies interfering with platelet function have been proposed in the literature, which could possibly have a positive impact on skin inflammation [6] such as antiplatelet drug P2Y12 antagonists, e.g., [24]. Another example is acetylsalicylic acid (commonly referred to under the trademark Aspirin), of which effects on psoriasis are poorly investigated but promising. It has anti-inflammatory properties [25] and positive effects on various immune disorders [26]. In addition, the identification of a proper and personalized diet could influence the development and progression of psoriasis, considering that the imbalance of gut microbiota and the deficiency of vitamin D or selenium are psoriasis-associated conditions [27].

Each growth factor examined in our experiments is now discussed.

First, TNF-α, which was released in high quantity on the first day of our experiments, is already recognized as a factor primarily involved in this pathology. It is produced by a variety of cells, such as keratinocytes, dendritic cells, and leukocytes. This implies that TNF-α might be a key factor in both the initial and chronic phases of psoriasis [28]. Still, TNF-α remains an enigmatic cytokine with respect to psoriasis pathogenesis because the anti-TNF biologics are not significantly effective in all psoriasis patients [29], suggesting that there may be differences in the inflammatory networks in skin lesions [30], perhaps driven by genetic background heterogeneity, with a different balance of protective and disease-associated alleles across several loci [31].

Second, VEGF [32] has a well-documented involvement in psoriasis, increasing angiogenesis and vascular permeability [33]. In addition, our data identified its increase before the onset of the symptoms. Wolf and Harrison [34] demonstrated that the psoriatic epidermis has a greater angiogenic activity associated with an increase in VEGF compared with the skin of normal subjects. Keratinocytes in lesioned skin are the major source of pro-angiogenic cytokines in psoriasis, including VEGF, TNF-α, and PDGF [35–37]. Moreover, VEGF increases in the plasma and serum of patients [33,38,39].

Third, TGF-β1 is considered a multipotent cytokine with a key role in many biological processes [40] among which is the pathogenesis of psoriasis [41]. To date, our data suggest a possible downregulation of TGF-β1 before the onset of symptoms, revealing an imbalance in the linked pathways, considering that it is involved in regulatory inflammation processes. TGF-β1 influences keratinocyte proliferation, activates angiogenesis, and promotes fibroblasts' production in the extracellular matrix. Data in the literature about psoriasis patients regard only studies on the established pathology, indicating an increase in this cytokine. For example, Meki et al. [33] reported that the serum levels of TGF-β1, in addition to VEGF and NO, were significantly higher in psoriasis patients than in controls and were sensitive to changes in disease severity. Moreover, mice overexpressing human TGF-β1 in epidermal keratinocytes develop a skin phenotype similar to human psoriasis [42]. Recently, it has been demonstrated that TGF-β1 mediates the development of psoriasis-like lesions via a Smad3-dependent, Th17-mediated mechanism [43].

Lastly, PDGF-AB assures steady contribution in the different phases of wound healing repair, which are angiogenesis, formation of fibrous tissue, and re-epithelialization [44]. Our data revealed an increase in PDGF-AB, suggesting an important role of this cytokine in the onset of psoriasis. In current literature, we have not found direct evidence of an increase in this growth factor in the plasma or serum of psoriasis patients. Nevertheless, the keratinocytes are directly involved in its production. Gu et al. [45] reported that, in vitro, the chemotactic and mitogenic responses of psoriatic fibroblasts to PDGF were significantly enhanced compared to normal fibroblasts. This increased sensitivity to PDGF could be related to the vascularization and inflammatory processes acting in the psoriatic dermis. Some years after, Krane et al. [46] observed a great increase in PDGF receptor expression in the fibroblast and bold vessels of the growth-activated dermis from chronic wounds and psoriatic lesions. They hypothesized that differential expression of PDGF receptors could regulate the increased proliferation of connective and vascular tissue cells observed in chronic wounds and psoriasis.

## 5. Conclusions

Taken together, these results show new aspects of this dermatological pathology, opening new possibilities both as a method of study, using a platelet-rich plasma preparation as CGF, and the involvement of platelets and their growth factors in their development and maintenance. Our data, obtained during the case study, should be validated in the future, and could address new clinical trials to better characterize the pattern expression of growth factors to identify new therapeutic targets.

**Author Contributions:** Substantial contributions to the conception or design of the work: E.B. and B.B.; acquisition, analysis, and interpretation of data for the work: E.B. and B.B.; drafting the work and revising it critically for important intellectual content: E.B., F.B. and R.R.; final approval of the version to be published: E.B., B.B., F.B. and R.R.; agreement to be accountable for all aspects of the work in ensuring that questions related to the accuracy or integrity of any part of the work are appropriately investigated and resolved: E.B. and R.R.; validation and bibliographic research, E.B., B.B. and F.B. All authors have read and agreed to the published version of the manuscript.

**Funding:** This research received no external funding.

**Institutional Review Board Statement:** The study was conducted in accordance with the Declaration of Helsinki. The authors are the owners of the samples.

**Informed Consent Statement:** Written informed consent for the re-use of human bio-specimens in scientific research has been obtained from the subjects to publish this paper.

**Data Availability Statement:** Data are contained within the article.

**Conflicts of Interest:** The authors declare no conflict of interest.

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
