# Peer review of "The Growth Factor Release from a Platelet-Rich Plasma Preparation Is Influenced by the Onset of Guttate Psoriasis: A Case Report"

_applsci, doi:10.3390/app12147250_

Round 1
Reviewer 1 Report
Dear Authors,
The authors only compared one patient and one healthy person, meaning that this case cannot be sufficient for their conclusion.
First of all, the authors provide insufficient data to conclude the relationship between 4 growth factors and guttate psoriasis.
The authors need to provide more scientific data.
Author Response
Dear Reviewer,
We revised the paper entitled “The growth factor release from a platelet-rich plasma preparation is influenced by the onset of guttate psoriasis: a case report.” to Applied Sciences (applsci-1766580).
We wish to thank you for your time and expertise. We are sorry for your opinion considering that this is a case report. Nevertheless, we implemented the introduction and discussion to improve the quality of the work.
Best regards.

Reviewer 2 Report
The present study by Borsani et al suggests the importance of Concentrated Growth Factors (CGF) from Platelet-rich plasma to study guttate psoriasis. They report a significant increase of growth factor release in the psoriasis subject with respect to the healthy control. While this pathology is unsurprising, the identification that the growth factors released from CGF were altered even before the onset of guttate psoriasis in the subject is of some interest.
Major comments:1. How pure is the CGF preparation to claim that the release of the growth factors is primarily from platelets and not from other cell types? This needs to be discussed comprehensively in the discussion. 2. The authors suggest in the discussion that "a new hypothesis could be made about the involvement of the platelets in these two pathologies and the CGF could be a useful tool for this kind of investigation". The role of platelet-related inflammation in the progression of Psoriasis has been reported earlier. Consequently, how CGF could provide an alternative for a better diagnostic as well as therapeutic intervention? A bit more detailed discussion on this in the present study would be helpful. 3. Though it is difficult to obtain a subject like V2 for an investigation like this. Still, it is difficult to give a conclusion like this from a single subject. This needs to be addressed in the discussion section. Minor comment: There are grammatical and spelling errors in the manuscript, which require corrections.
Author Response
Dear Reviewer,
We revised the paper entitled “The growth factor release from a platelet-rich plasma preparation is influenced by the onset of guttate psoriasis: a case report.” to Applied Sciences (applsci-1766580).
We wish to thank you for your time and expertise. We have attempted to address each concern with either additional information and/or clarifications within the text. The critiques proved very helpful, and we feel that in addressing these concerns, we have improved the quality of our study. Below, we handle each critique and specifically detail where in the revised manuscript we addressed the point. Moreover, we marked in red color the corrections in the revised manuscript.
The present study by Borsani et al suggests the importance of Concentrated Growth Factors (CGF) from Platelet-rich plasma to study guttate psoriasis. They report a significant increase of growth factor release in the psoriasis subject with respect to the healthy control. While this pathology is unsurprising, the identification that the growth factors released from CGF were altered even before the onset of guttate psoriasis in the subject is of some interest.
Major comments:
- How pure is the CGF preparation to claim that the release of the growth factors is primarily from platelets and not from other cell types? This needs to be discussed comprehensively in the discussion.
REPLY: Thank you for your precious observation. The following paragraph has been added to the Discussion (page 5, line 153).
“So, the data obtained cannot be associated with only one cell type even if in this study, based on our previous morphological research, the contribution of platelets in the growth factor production was expected to be important [14,19]. The quantitative count of the different cell types was hindered by the solid nature of CGF, in fact, the cells are entrapped in the fibrin network formed during the centrifugation without the aid of additives. Nevertheless, a study by Anitua et al. [16] evidenced a specific trend of VEGF release and an alteration of the fibrin network in presence of leukocytes. Two different platelet-rich plasma preparations have been considered: leucocyte-free and leucocyte-rich. The release of different growth factors was evaluated in vitro over 8 days and the availability of VEGF greatly decreased after 3 days and was almost absent on the 7th day using the leucocyte-rich preparation. On the contrary, the VEGF kinetic release observed in our experiments on CGF was similar to leucocyte-free preparation. Moreover, the fibrin network in the leukocyte-rich preparation resulted in a more heterogeneous and loose mesh, which was less evident in CGF during our previous studies [14].
- The authors suggest in the discussion that "a new hypothesis could be made about the involvement of the platelets in these two pathologies and the CGF could be a useful tool for this kind of investigation". The role of platelet-related inflammation in the progression of Psoriasis has been reported earlier. Consequently, how CGF could provide an alternative for a better diagnostic as well as therapeutic intervention? A bit more detailed discussion on this in the present study would be helpful.
REPLY: Thank you for your suggestion, we think that the discussion has been implemented. The following sentence has been modified in the Discussion (page 4, line 14).
“If the finding of this study could be confirmed by a wide clinical study, a new hypothesis could be made about the involvement of the platelets in the onset of these two pathologies this pathology and the CGF could be a useful in vitrotool for this kind of investigation to evaluate the kinetic release of growth factors over a period of time, moreover, it could simulate the psoriasis microenvironment in 2D and 3D human skin models.”
- Though it is difficult to obtain a subject like V2 for an investigation like this. Still, it is difficult to give a conclusion like this from a single subject. This needs to be addressed in the discussion section.
REPLY: Thank you for your suggestion.
The following paragraph has been modified in Discussion (page 4 – line 147):
“Nevertheless, this study presents some limitations which must be considered in the comprehensive evaluation of the obtained results during the case study. The first limitation is the evaluation of only one psoriasis subject, that is a case report. So, the results want to address to new hypothesis on the onset of this pathology considering the precious data of this study which should be validated in the future. The second limitation is the presence of leucocytes in CGF, which could secrete TNF-α, VEGF, TGF- β1, and PDGF-AB. So, the data obtained cannot be associated with only one cell type…….”
The following sentence has been modified in Conclusions (page 6 – line 246):
“Our data, obtained during the case study, should be validated in the future, and could address new clinical studies trialsto better characterize the pattern expression of growth factors to identify new therapeutic targets.
Minor comment: There are grammatical and spelling errors in the manuscript, which require corrections.
REPLY: The text has been revised for the English language.

Reviewer 3 Report
The authors through their case-study are evaluating the possibility of concentrated growth factors and their released cytokines as a method of diagnostics/studying systemic pathologies such as psoraisis.
The authors have indicated that cytokines such as PDGF, VEGF, TNFa are increased in psorasis, indicating an increase in angiogenesis and inflammation. However, TGFB is reduced. The authors have attributed this finding to an imbalance of inflammatory pathways. The authors also clearly highlight the limitations of the study that these factors cannot be attributed to one cell type.
Their data is important because they reflect the growth factor status before the onset of disease.
Therefore, I have no further comments against the manuscript.
Author Response
Dear Reviewer,
We revised the paper entitled “The growth factor release from a platelet-rich plasma preparation is influenced by the onset of guttate psoriasis: a case report.” to Applied Sciences (applsci-1766580).
We wish to thank you for your time, expertise, and positive opinion of our work.
Best regards

Reviewer 4 Report
The authors have provided a well-presented case report providing new evidence that platelet have dysregulated release of growth factors prior to a psoriasis onset. Measurement of such factors as part of routine blood work may allow implementation of preventive therapies to alleviate psoriasis development and ameliorate patient quality of life. The manuscript is clearly written and data are well presented. Minor comments are the following:
- Line 80-85: since blood is collected in a serum tube, platelet activation and release of factors is likely to happen in the collection tube. Hence, further release of factors in the culture system likely depends on immune cells. Throughout the manuscripts, the authors should revise the introduction (Line 36-38) to explain how and when immune cells, and not only platelets, release the reported growth factors. This is nicely reviewed by the authors in the discussion.
- In the introduction and in line 150-152, the authors mention that MVP, a commonly measured parameters of blood work, may be used as marker of platelet activation. If available, such value should be shown for subject V1 and V2 in the figure(s), and comments should be made on the value of MVP as an alternative prognostic value for psoriasis development. In addition, and if available from the blood work analysis, the concentration of different immune cells in subjects V1 and V2 should be reported and comments should be made on the most likely source of the reported growth factors, and whether leukocyte levels might represent another prognostic biomarker of psoriasis development.
- In the discussion, the authors should describe potential preventive intervention that could be provided to patients before psoriasis onset in the case that biomarkers here identified allow early patient identification.
Author Response
Dear Reviewer,
We revised the paper entitled “The growth factor release from a platelet-rich plasma preparation is influenced by the onset of guttate psoriasis: a case report.” to Applied Sciences (applsci-1766580).
We wish to thank you for your time and expertise. We have attempted to address each concern with either additional information and/or clarifications within the text. The critiques proved very helpful, and we feel that in addressing these concerns, we have improved the quality of our study. Below, we handle each critique and specifically detail where in the revised manuscript we addressed the point. Moreover, we marked in red color the corrections in the revised manuscript.
The authors have provided a well-presented case report providing new evidence that platelet have dysregulated release of growth factors prior to a psoriasis onset.
Measurement of such factors as part of routine blood work may allow implementation of preventive therapies to alleviate psoriasis development and ameliorate patient quality of life. The manuscript is clearly written and data are well presented.
Minor comments are the following:
- Line 80-85: since blood is collected in a serum tube, platelet activation and release of factors is likely to happen in the collection tube. Hence, further release of factors in the culture system likely depends on immune cells. Throughout the manuscripts, the authors should revise the introduction (Line 36-38) to explain how and when immune cells, and not only platelets, release the reported growth factors. This is nicely reviewed by the authors in the discussion.
REPLY: Thank you for your suggestion.
The following sentence has been added to the Introduction (page 2, line 40).
“…nevertheless, they are able also to synthesize the growth factors up to over 7 days after activation from accumulated mRNA [7].”
Moreover, the following paragraph has been added to the Introduction (page 2, line 52).
“Our previous data [14] showed a specific kinetic of accumulation for each growth factor analyzed in vitro over 8 days, sustaining the ability of platelets to guarantee the release of the growth factors for several days after activation [7]. The leukocytes contribute to growth factor accumulation as reported for other platelet-rich preparations [e.g., 16].”
- In the introduction and in line 150-152, the authors mention that MVP, a commonly measured parameters of blood work, may be used as marker of platelet activation. If available, such value should be shown for subject V1 and V2 in the figure(s), and comments should be made on the value of MVP as an alternative prognostic value for psoriasis development.
REPLY: We totally agree with your observation, but the data are not available. We are very sorry, but it is necessary to consider that the obtained data on the psoriasis subject was unexpected.
In addition, and if available from the blood work analysis, the concentration of different immune cells in subjects V1 and V2 should be reported and comments should be made on the most likely source of the reported growth factors, and whether leukocyte levels might represent another prognostic biomarker of psoriasis development.
REPLY: We totally agree with your observation, but the data are not available. We are very sorry, but it is necessary to consider that the obtained data on the psoriasis subject was unexpected.
- In the discussion, the authors should describe potential preventive intervention that could be provided to patients before psoriasis onset in the case that biomarkers here identified allow early patient identification.
REPLY: Thank you for your interesting suggestion, the following paragraph has been added to the Discussion (page 5, line 184).
“In this context, a clinical relapse of this preliminary report could introduce the idea of a potential preventive intervention that could be provided to patients before psoriasis onset with the aim to modulate growth factor secretion. Some target pharmacological therapies interfering with platelet functions have been proposed in the literature, whichcould possibly have a positive impact on skin inflammation [6] such as antiplatelet drug P2Y12 antagonists [e.g., 24].Another example is the acetylsalicylic acid (commonly referred to under the trademark Aspirin), which effects on psoriasis are poorly investigated but promising. Indeed, it has anti-inflammatory properties [25] and positive effects on various immune disorders [26]. In addition, the identification of a proper and personalized diet could influence the development and progression of psoriasis, considering that the imbalance of gut microbiota and the deficiency of vitamin D or selenium are psoriasis-associated conditions [27].”

Round 2
Reviewer 1 Report
Dear Authors
Although this manuscript is a case report, the number of participants is only one, and too small.
This revised manuscript is much bettern than before.